# Soil Organic Carbon Prediction Based on Different Combinations of Hyperspectral Feature Selection and Regression Algorithms

**Naijie Chang** [1],[†] , **Xiaowen Jing** [1],[†], **Wenlong Zeng** [2], **Yungui Zhang** [1], **Zhihong Li** [1], **Di Chen** [3], **Daibing Jiang** [2], **Xiaoli Zhong** [2], **Guiquan Dong** [2] and **Qingli Liu** [1],*

[1] State Key Laboratory of Efficient Utilization of Arid and Semi-Arid Arable Land in Northern China, Institute of Agricultural Resources and Regional Planning, Chinese Academy of Agricultural Sciences, Beijing 100081, China; changnaijie@gmail.com (N.C.)

[2] Longyan Branch of Fujian Tobacco Research Institute, Longyan 364000, China

[3] Institute of Environment and Sustainable Development in Agriculture, Chinese Academy of Agricultural Sciences, Beijing 100081, China

* Correspondence: liuqingli@caas.cn

[†] These authors contributed equally to this work.

**Abstract:** Cropland soil organic carbon (SOC) is crucial for global food security and mitigating the greenhouse effect. Accurate SOC prediction using hyperspectral data is essential for dynamic monitoring of soil carbon pools in croplands. However, effective methods to reduce hyperspectral data dimensionality and integrate it with suitable regression algorithms for reliable prediction models are poorly understood. In this study, we analyzed 108 soil samples from Changting County, Fujian Province, China. Our objective was to evaluate the performance of various combinations of six feature selection methods and four regression algorithms for SOC prediction. Our findings are as follows: the combination of the Successive Projections Algorithm (SPA) and Partial Least Squares (PLS) yielded the most favorable results, with $R^2$ (0.61), RMSE (1.77 g/kg), and MAE (1.48 g/kg). Moreover, we determined the relative importance of variables, with the following ranking: 696 nm > 892 nm > 783 nm > 1641 nm > 1436 nm > 396 nm > 392 nm > 2239 nm > 2129 nm. Notably, 696 nm exhibited the highest importance in the SPA-PLS model, with the Variable Importance in Projection (VIP) value of 1.22. This study provides profound insights into feature selection methods and regression algorithms for SOC prediction, highlighting the superiority of SPA-PLS as the optimal combination.

**Keywords:** soil organic carbon; hyperspectral; feature selection; regression; relative importance



## 1. Introduction

Soils represent the largest terrestrial carbon reservoir and are abundant in organic carbon. Croplands, in particular, contain substantial carbon stocks and play a crucial role in the terrestrial carbon cycle, contributing to the maintenance of the global carbon balance and food security [1,2]. Unlike other terrestrial ecosystems, SOC in croplands is highly sensitive to agricultural management practices, in addition to ecological factors. This interplay of influences results in the spatial heterogeneity of SOC in croplands, with implications for both the global carbon cycle and food production. Thus, timely monitoring of SOC content is essential for effective crop growth management, soil carbon sequestration, soil resource management, and the promotion of sustainable agricultural development.

The acquisition of SOC content is often contingent upon the collection of a substantial volume of measurement data. However, this process may be impeded by various factors, including the sampling environment, the costs associated with data acquisition, and the need for accuracy in measurements [3]. Some conventional methods, by means of the digestion, oxidation, and titration of agents for determining SOC content, have been shown

to generate non-recyclable toxic waste (such as potassium dichromate and sulfuric acid), which poses a significant threat to the environment [4]. The environmental pollution caused by these methods is a major concern for researchers and policymakers alike. Therefore, there is an urgent need to develop alternative methods that are both accurate and environmentally friendly. In light of these concerns, spectral technology has emerged as a promising alternative, offering advantages such as low cost and portability for the measurement of SOC content. Visible near-infrared (VIS-NIR) hyperspectral has gained widespread use in the SOC spectral inversion due to its rapid, non-destructive, non-polluting, and cost-effective nature [5]. Numerous studies have demonstrated a significant inverse correlation between the soil spectral reflectance and SOC content [6,7], and have shown that VIS-NIR data (350–2500 nm) are effective in predicting SOC content [8]. However, the high dimensionality of hyperspectral data, comprising hundreds or thousands of spectral bands, presents computational challenges for analysis and modeling. To address this issue, feature selection techniques can be employed to extract the most informative spectral features and reduce data dimensionality.

Several feature selection methods have been proposed to extract meaningful and informative features from hyperspectral data [8]. Among these methods, mutual information is a versatile technique that measures the shared information between variables and captures both linear and nonlinear relationships in high-dimensional soil hyperspectrum datasets [9]. However, computational demands must be considered for datasets with numerous dimensions. While the correlation coefficient is a straightforward metric for assessing linear associations between hyperspectral bands, it may fall short in capturing complex nonlinear relationships [10]. The Successive Projections Algorithm (SPA) preserves information by iteratively selecting relevant features, but faces challenges in computational complexity and potential oversight of global interactions [11]. Competitive Adaptive Reweighted Sampling (CARS) presents a promising nonlinear feature selection technique that incorporates both local and global feature interactions. However, optimal performance relies on careful parameter tuning, and scalability can be an issue [12]. Machine learning algorithms, such as the Random Forest algorithm, have also been employed to perform feature selection of hyperspectral data by leveraging decision tree ensembles. However, interpretability and overfitting risks should be considered [13]. Therefore, a comparison of different feature selection methods is required when reducing the spectral data to obtain the optimal features selection.

Multivariate statistical techniques are commonly employed for the analysis of soil spectral data. The effectiveness of these methods relies on the chosen regression algorithm [14]. Among linear approaches, PLS stands out as the most widely used method for elucidating the connection between spectral data and soil properties, owing to its interpretability and low computational requirements [15]. However, the relationship between spectral data and soil properties is not always linear, which may limit the suitability of PLS for modeling soil properties [16]. In light of this concern, several studies have explored the utilization of machine learning techniques to achieve higher accuracy compared to linear regression [17,18]. For instance, de Santana et al. [19] conducted a comparative analysis between Random Forest (RF) and PLS and observed that RF exhibited a slight improvement over PLS, while also enabling the identification of outliers through a proximity matrix. In one of the pioneering investigations, Padarian et al. [20] demonstrated that Convolutional Neural Networks (CNNs) outperformed both PLS and Cubist models in a topsoil dataset from Europe. In contrast, Nawar et al. [21] reported the superior performance of PLS over Support Vector Machine Regression (SVR) specifically for salt-affected soils. When predicting spectral data by soil organic carbon regression, it is necessary to choose an appropriate regression algorithm based on specific soil conditions.

The integration of feature selection techniques with regression models has been shown to enhance the accuracy of SOC prediction, while simultaneously reducing model complexity [22]. Raj et al. [23] employed a combination of a correlation coefficient and variable importance in projection with PLS and SVR, and discovered that the use of a mutual

information indicator in conjunction with SVR yielded an $R^2$ value of 0.68, outperforming the full-spectrum PLSR model. Despite these promising results, comprehensive evaluations of multiple feature selection and modeling method combinations for SOC prediction remain limited [24]. According to the characteristics of cropland soil in different regions, the combination and comparison of different feature selection and regression algorithms are crucial to the accurate acquisition of SOC.

In this study, we evaluated different combinations of feature selection methods, including correlation coefficient (R), Mutual Information (MI), CARS, SPA, and RF, and regression algorithms, including RF, Gradient Boosting Regression (GBR), SVR, and PLS, for SOC prediction based on soil hyperspectral data. Our objectives were twofold: (a) to investigate the performance of feature selection methods and regression algorithms and identify the optimal algorithm combination for SOC prediction; and (b) to construct an SOC prediction model based on the optimal algorithm combination, derive a quantitative formula for SOC, and analyze the relative importance of different sensitive bands.

## 2. Materials and Methods

### 2.1. Study Area

The study area is located in Changting county, Fujian province, China (116.294° E–116.299° E, 25.550° N–25.557° N) and is characterized by a high terrain in the north and low terrain in the south with an average elevation of 251 m. The region experiences a middle subtropical monsoon climate, with an annual mean temperature of 18.5 °C, a frost-free period of 260 days, total precipitation of 1712 mm, and an average relative humidity of 80%. The predominant soil is silt loam, as classified by the American system of soil texture classification. The land is primarily used for cropland, with rice, sweet potatoes, and flue-cured tobacco being the main crops grown. Figure 1 provides a detailed geographical distribution of the study area.

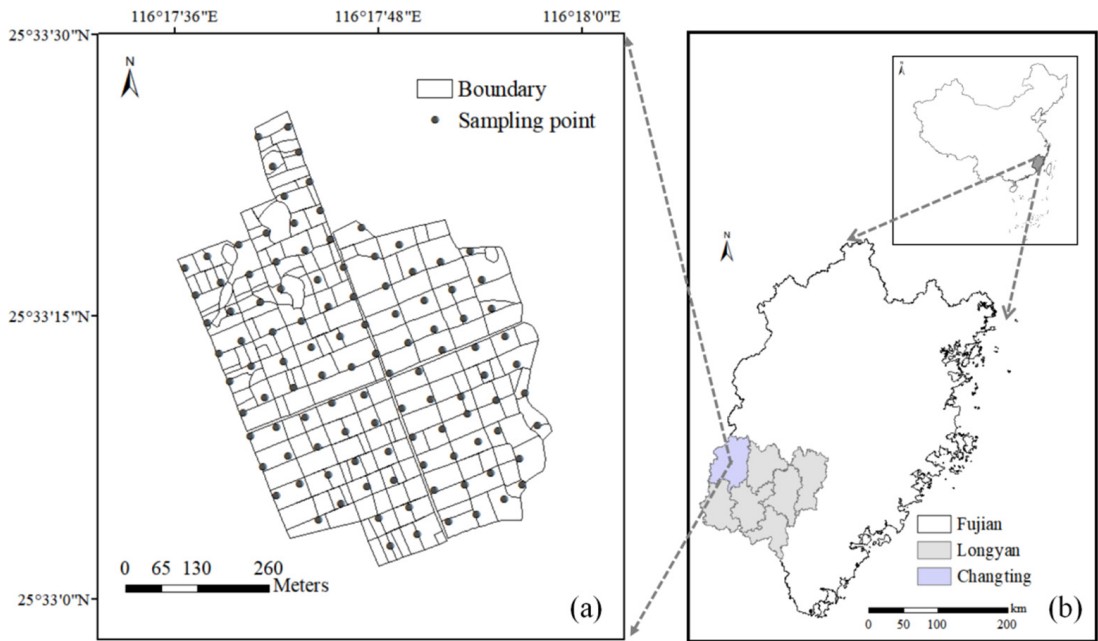

**Figure 1.** Study area and soil samples spatial distribution. (**a**) Location of the study area. (**b**) The spatial distribution of sampling points and the boundaries of farmland.

### 2.2. Soil Sample Collection and Chemical Analysis

In February 2021, a comprehensive soil sampling campaign was conducted within a designated study area. A total of 108 soil samples were carefully collected from the uppermost 20 cm layer of soil at spatially distributed 50 m × 50 m grid sampling points (refer to Figure 1). The positioning of the sampling points was accurately recorded using a handheld

global positioning instrument (GPS, 59222-C10), which exhibited a positioning error within the range of 1 m or less. To ensure the representativeness of the collected samples, careful considerations were given to the selection of suitable sampling locations. Specifically, areas with minimal vegetation and other potential disturbances were identified, allowing for a clear and undisturbed soil surface. Each soil sample was conscientiously obtained by compositing 5 individual soil sub-samples, which were collected within a circular area with a radius of 5 m at the center of each grid square. Upon collection, the soil samples were subjected to meticulous laboratory procedures. Firstly, rocks and any visible plant residues were carefully removed from the samples. Subsequently, the samples were subjected to air drying, followed by sieving at a particle size of 2 mm. This standard sieving procedure ensured the removal of larger debris and facilitated consistent sample preparation across all collected soil samples. The collected soil samples were then divided into two distinct portions for subsequent analyses. The first portion was dedicated to the determination of soil spectral reflectance, as detailed in Section 2.3 of this study. The second portion was allocated for the quantification of SOC content, utilizing the well-established Walkley and Black method [25]. By adhering to these standardized procedures, we ensured the collection of high-quality soil samples, thereby establishing a solid foundation for subsequent analyses and investigations.

### 2.3. Hyperspectral Data Acquisition

The collection of soil spectral data was conducted using a GaiaSorter-dual dual-camera full-spectrum hyperspectral sorter (Dualix Instruments Co., Ltd., Chengdu, China) within a controlled darkroom environment. This instrument possessed a notable spectral acquisition range spanning from 363 to 2583 nm, with a spectral resolution of 2–3 nm (360–1009 nm) and 5–6 nm (1011–2500 nm). During the experimental setup, the soil samples were carefully positioned in glass dishes, featuring a uniform depth of 1.5 cm and a radius of 5 cm, according to the designated order indicated by their respective labels. Subsequently, the soil surfaces were carefully leveled, using a ruler to ensure consistent sample preparation. As for the light source, a 50 W halogen lamp was employed, positioned at a precise top angle of $15°$ and a field of view of $8°$ relative to the soil sample surface. The camera probe, positioned vertically at a distance of 30 cm above the soil sample, was employed to capture the spectral information. Prior to initiating the data collection process, necessary calibration procedures were conducted, including whiteboard correction, to maintain the accuracy of the instrument. Adequate stabilization time was allotted to the instrument to ensure optimal performance before commencing with the spectral data acquisition. To obtain representative spectral reflection data for each soil sample, a defined region of interest (ROI) was precisely demarcated as a rectangular area spanning $60 \times 60$ pixels within the acquired image. The spectral curves within the designated ROIs were then extracted, followed by an arithmetic averaging process, resulting in the determination of the actual spectral reflection data for each individual soil sample. By adhering to these rigorous procedures, we ensured the acquisition of precise and reliable soil spectral data, thereby establishing a robust foundation for subsequent analyses and interpretations.

### 2.4. Feature Selection Algorithms

In this study, three types of feature selection methods were employed to address the research question: traditional statistical methods, classical spectral feature selection methods, and machine learning methods.

For traditional statistical methods, the MI and R methods were selected. MI measures the dependence between two variables and can be used to quantify the relevance of a feature by measuring its mutual information with the target variable [26]. High MI values indicate strong dependence and suggest that the feature contains valuable information for predicting the target variable. The correlation method calculates the correlation coefficient between each feature and the target variable, with larger absolute values indicating stronger correlation and greater relevance [10]. While these methods can be effective in reducing

data dimensionality and selecting relevant features, they may not always capture complex relationships between features and the target variable.

For classical spectral feature selection methods, the SPA and CARS methods were chosen. SPA is a powerful tool for selecting a small set of representative spectral variables with minimal redundancy and an emphasis on minimizing collinearity [27]. It solves quadratic optimization problems subject to linear equality and inequality restrictions. CARS, on the other hand, is a variable selection method for data with high correlation and for scenarios where the number of variables greatly exceeds the number of samples. It selects an optimal combination of key wavelengths from multi-component spectral data based on the principle of 'survival of the fittest' from Darwin's Evolution Theory [28].

For machine learning, the RF algorithm was utilized in this study. RF consists of multiple decision trees, with each node representing a condition on a feature designed to split the dataset based on different response variables [29]. Impurity measures, such as variance or least squares fitting, are typically employed for regression tasks to determine the optimal condition at each node. During training, the reduction in impurity attributable to each feature can be calculated and used as a metric for feature selection. The average reduction in impurity for each feature can be calculated for a RF.

To compare the effectiveness of different feature selection algorithms, the raw data (RAW) were used as a baseline in the feature selection analysis.

*2.5. SOC Regression Algorithms*

The spectral data with feature selection were used as input variables for the SOC estimation regression model. In this study, several model building algorithms were employed, including PLS, RF, GBR, and SVR.

PLS is a statistical technique that shares similarities with Principal Components Regression. Rather than identifying hyperplanes of maximum variance between the response and independent variables, PLS constructs a linear regression model by projecting both the predicted and observable variables onto a new space [7]. It is well-suited to scenarios where both sets of variables exhibit multicollinearity and have many columns, and where the number of observed data samples is relatively small. The parameters of PLS in this study were configured to their default values, as defined in the Python Sklearn package.

RF is an ensemble learning algorithm that integrates multiple decision trees to reduce the risk of overfitting. The training process of these decision trees occurs in parallel, with slight variations introduced through a randomization process within the algorithm. This results in a robust model capable of accurately predicting outcomes [30]. In RF modeling, three parameters need to be specified: the number of trees in the forest (n_estimators), the minimum number of samples required to be at a leaf node (min_samples_leaf), and the minimum number of samples required to split an internal node (min_samples_split). In this study, these parameters were set to 100, 1, and 2, respectively.

GBR represents an advancement of the boosting technique, enabling its application to diverse differentiable loss functions [31]. The fundamental principle of GBR entails the iterative creation of an additive model in a forward stagewise manner, thereby facilitating the optimization of arbitrary differentiable loss functions. In each stage, a regression tree is precisely fitted to the negative gradient of the specific loss function under consideration. By leveraging the stochastic gradient boosting procedure, the overfitting phenomenon can be mitigated, consequently leading to enhanced model performance. GBR inherently possesses the capability to effectively handle mixed-type data, thereby endowing it with formidable predictive abilities. Notably, GBR exhibits exceptional resilience to outliers in the output space, courtesy of its robust loss functions. In GBR modeling, six parameters are defined by the user: the loss function (loss), the number of boosting stages to perform (n_estimators), the minimum number of samples required to split an internal node (min_samples_split), the minimum number of samples required to be at a leaf node (min_samples_leaf), the learning rate (learning_rate), and the maximum depth of the individual regression estimators (max_depth). In this study, these parameters

were specified as follows: loss = "squared_error", learning_rate = 0.1, n_estimators = 100, min_samples_split = 2, min_samples_leaf = 1, and max_depth = 3.

The SVR algorithm is a supervised machine learning technique with applications in both classification and regression tasks [32]. The algorithm identifies the optimal hyperplane that separates data into distinct classes or predicts the target variable. In classification tasks, this hyperplane is selected to maximize the margin between classes, defined as the distance between the hyperplane and the nearest data points from each class. These nearest data points are known as support vectors, giving rise to the name of the algorithm. SVM can handle both linearly and non-linearly separable data through the use of kernel functions that map data into higher-dimensional space, where a linear hyperplane can be employed to separate classes or predict the target variable. In SVR modeling, the kernel type (kernel), Degree of the polynomial kernel (degree), Regularization parameter (C), and Epsilon in the epsilon-SVR model (epsilon) need to be specified. In this study, these parameters were set to kernel = "rbf", degree = 3, C = 1.0, and epsilon = 0.1, respectively.

### 2.6. Statistical Modeling and Accuracy Assessment

To optimize the selection of SOC prediction model methods, this study combined 6 feature selection methods, including raw soil reflectance data, with 4 machine learning regression algorithms, resulting in 24 combinations. The performance of each combination was assessed using the coefficient of determination ($R^2$), root-mean-square error (RMSE), and mean absolute error (MAE). The dataset was split into a 70/30 ratio for model construction and validation, respectively. The predictive efficacy of the model was evaluated using the aforementioned metrics. The formulas for these metrics are provided below:

$$R^2 = 1 - \frac{\sum_{i=1}^{n}(\hat{y}_i - \overline{y})^2}{\sum_{i=1}^{n}(y_i - \overline{y})^2} \tag{1}$$

$$RMSE = \sqrt{\frac{1}{n}\sum_{i=1}^{n}(y_i - \hat{y}_i)^2} \tag{2}$$

$$MAE = \frac{1}{n}\sum_{i=1}^{n}|\hat{y}_i - y_i| \tag{3}$$

where $n$ represent the number of soil samples, $y_i$ and $\hat{y}_i$ are the observed SOC and the predicted SOC, respectively. $\overline{y}$ is the average observation.

In this study, statistical analyses (Sklearn and Statsmodels packages) and graphical (Matplotlib and Seaborn packages) representations were conducted using the Python programming language (version 3.9.1).

## 3. Results

### 3.1. Description of SOC Content and Hyperspectral Characteristics

SOC content ranged from 8.7 to 24.5 g/kg, with a mean value of 17.02 g/kg (SD = 3.24 g/kg). The data distribution was found to be normal, according to the Shapiro–Wilk test. Kernel density estimation analysis revealed that the probabilities of SOC falling within the ranges of 7.29–26.75 g/kg (Mean ± 3 × SD), 10.53–23.51 g/kg (Mean ± 2 × SD), and 13.78–20.26 g/kg (Mean ± 1 × SD) were 100%, 94%, and 64%, respectively (Figure 2a).

The hyperspectral characteristics of soil samples are presented in Figure 2b. A significant linear increasing trend (slope of $3 \times 10^{-4}$, $R^2 = 0.92$, $p < 0.001$) was observed between soil reflectance and wavelength within the range of 428–1763 nm. The soil reflectance curve was relatively flat within the range of 1769–2533 nm. Below 2328 nm, the 95% confidence interval for soil reflectance was closer to the average line, indicating that soil reflectivity was more consistent among different soil samples.

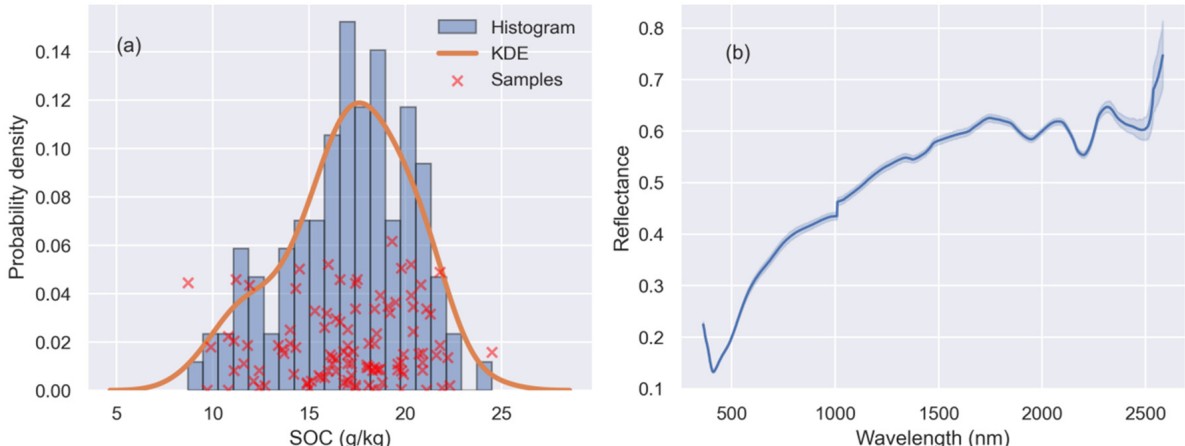

**Figure 2.** SOC content and spectral characteristics description of soil samples. (**a**) Kernel density estimation curve and probability distribution of SOC. (**b**) Hyperspectral characteristics of soil samples; the blue line represents average reflectance of all soil samples, the light blue is 95% confidence of data.

### 3.2. Feature Selection of Soil Spectral Data

To improve the precision of hyperspectral data modeling and minimize redundancy, we employed five feature selection techniques in the feature selection process (Figure 3). For conventional statistical and machine learning approaches, we selected the top 10 bands based on their correlation coefficient, mutual information value, and relative importance between spectral reflectance and SOC using the R, MI, and RF methods, respectively. With the R method, all selected bands exhibited a significant negative correlation with SOC ($p < 0.001$) and coefficients below $-0.54$. The coefficients of 612–617 nm were higher among selected bands. With the MI method, all mutual information values were above 0.281. The mutual information regression values of 612 nm (0.293), 615 nm (0.292), and 592 nm (0.289) were relatively higher among selected bands. With the RF method, the relative importance of bands was as follows: 401 nm > 413 nm > 1564 nm > 2583 nm > 404 nm > 543 nm > 558 nm > 563 nm > 565 nm > 425 nm. This RF-based feature selection process model produced RMSE and MAE values of 0.89 g/kg and 0.67 g/kg, respectively.

SPA employs vector projection analysis to compare projection vector sizes by projecting them onto other wavelengths. The wavelength with the largest projection vector is selected based on a correction model. SPA selects a combination of variables with minimal collinearity and the least redundant information. During the selection process, the RMSECV of different variable sets exhibited a trend of initial decrease, followed by stabilization. When the number of variables was 9, the RMSECV was significantly lower than the minimum RMSECV when the number of variables was greater than 9 ($p < 0.05$, F-test). The selected bands were 783 nm, 696 nm, 2129 nm, 396 nm, 2239 nm, 892 nm, 1641 nm, 1436 nm, and 392 nm, with an RMSECV of 2.22 g/kg.

CARS selects wavelength points with large absolute values of regression coefficients in the PLS model through adaptive reweighted sampling technology. Wavelength points with small weights are removed, and the subset with the lowest RMSECV is selected through interactive verification, effectively identifying the optimal variable combination. The RMSECV exhibited a trend of initial slow decline, followed by a rise. When the number of Monte-Carlo iterations was 50, the CARS method selected 39 bands, yielding a minimum cross-validated RMSE of 1.35 g/kg. Original soil reflectance data without feature selection were also included as input variables in the SOC prediction model process.

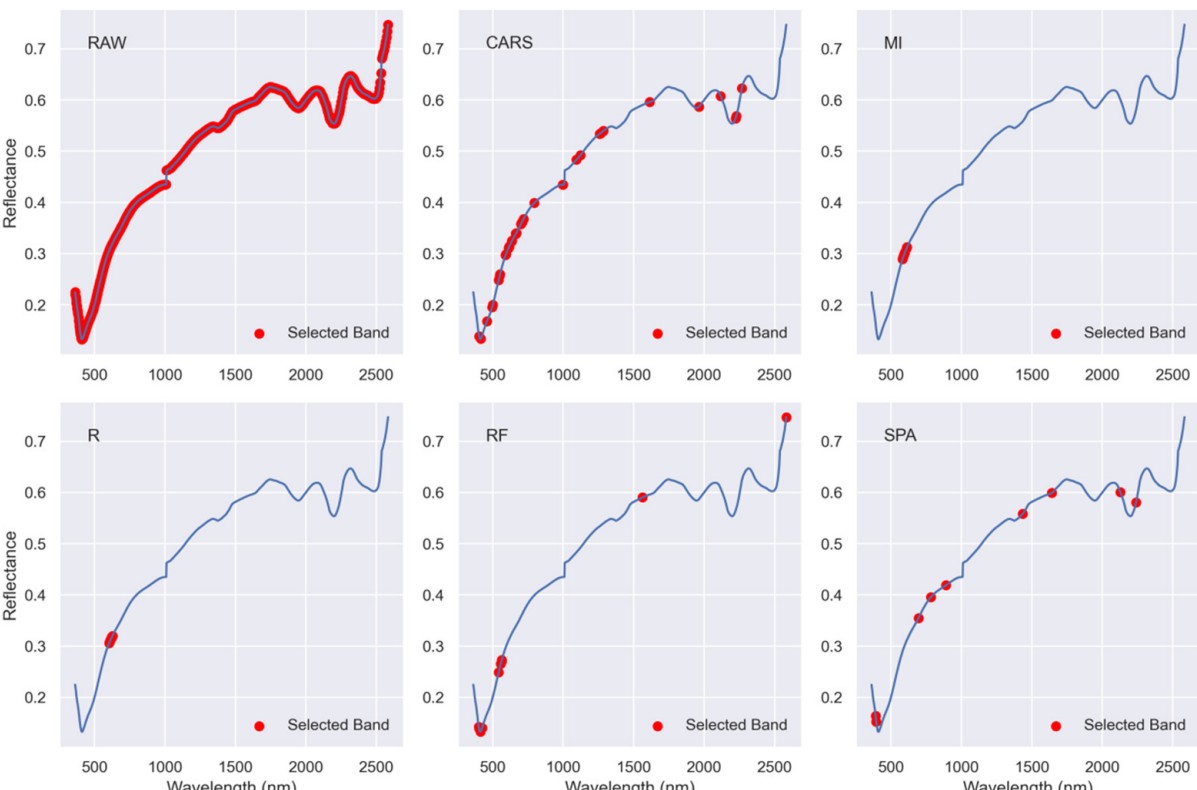

**Figure 3.** Process for the characteristic spectral bands selection. RAW is the original soil reflectance data, all bands were selected; CARS is the Competitive Adaptive Reweighted Sampling, and 39 bands were selected; MI, R, and RF are the Mutual Information, Correlation Coefficient, and Random Forest methods, respectively, and the final number of selected bands was 10 for each method; SPA is Successive Projections Algorithm, and 9 bands were selected.

### 3.3. SOC Prediction Model Construction

Figure 4 illustrates the comparative performance of multiple algorithm combinations on a test dataset. In contrast to RAW, combinations employing the MI, R, CARS, and RF methods exhibited diminished $R^2$ values and elevated RMSE and MAE values overall. This indicates that these algorithms may have inadvertently discarded pertinent sample information during feature selection, resulting in suboptimal model performance. SPA demonstrated robust performance overall; in comparison to the optimal combination in RAW, the $R^2$ value for the best SPA combination increased by 17.31%, while RMSE and MAE decreased by 9.69% and 1.33%, respectively. Among SOC regression algorithms, PLS and RF exhibited significantly higher $R^2$ values ($p < 0.05$, *t*-test) and significantly lower RMSE and MAE values ($p < 0.05$, *t*-test) than the GBR and SVR methods. Although PLS did not differ significantly from RF in terms of $R^2$, RMSE, and MAE values, the optimal PLS combination algorithm improved upon the best RF $R^2$ value by 19.61%, while RMSE and MAE decreased by 10.61% and 1.33%, respectively.

Figure 5 depicts the top six algorithm combinations, ranked according to their $R^2$, RMSE, and MAE values. In descending order of performance, the algorithm combinations are SPA_PLS, RAW-PLS, RAW-RF, RF_RF, CARS_RF, and RF_PLS. Among feature selection methods, CARS, SPA, and RF were included; among SOC regression algorithms, only RF and PLS were included. The SPA-PLS combination yielded the highest $R^2$ value (0.61) and the lowest RMSE (1.77 g/kg) and MAE (1.48 g/kg), indicating that it was the optimal algorithm combination for predicting SOC.

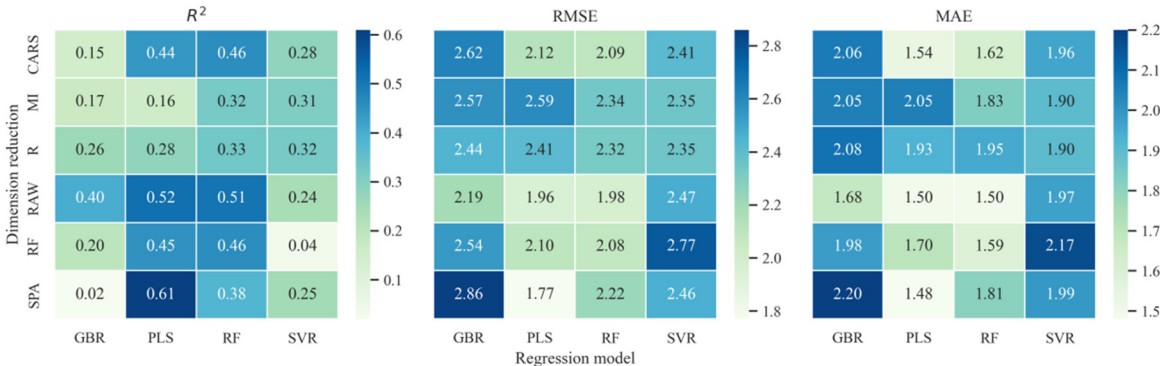

**Figure 4.** Comparison of the combination of different feature selection and SOC modeling methods. The performance of the SOC prediction model was assessed using $R^2$, RMSE, and MAE. A value closer to 0 for $R^2$ indicates better model performance, while for RMSE and MAE, values closer to 0 indicate a more accurate model.

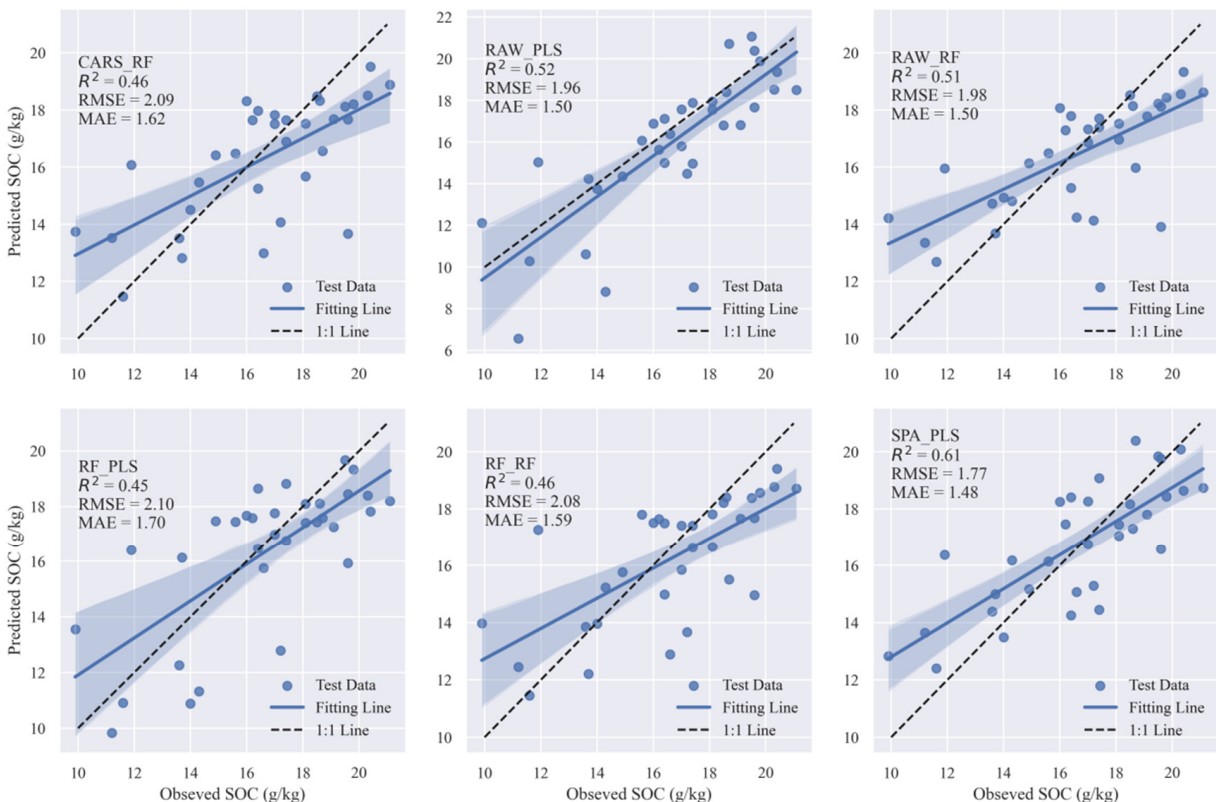

**Figure 5.** The top six combinations of feature selection and SOC modeling methods were evaluated based on $R^2$, RMSE, and MAE. The 1:1 line of observed data is represented by a black dashed line, while the regression fit line between the observed and predicted SOC is depicted in blue.

To evaluate the relative importance of different bands in the optimization algorithm combination, we calculated the VIP values of the SPA-PLS-based SOC prediction model and presented the results in Figure 6. As shown in Figure 6, the VIP values for wavelengths 696 nm, 892 nm, 783 nm, 1641 nm, and 1436 nm exceeded a value of 1, indicating their higher relative importance. Among these wavelengths, 696 nm exhibited the highest VIP value at 1.22, suggesting that this band plays a crucial role in the SPA-PLS model. Further-

more, SOC can be computed using Equation (4) based on the SPA-PLS SOC prediction model (Equation (4)).

$$
\begin{aligned}
Ysoc = {}& 32.59 + 510.817 \times W783 - 582.416 \times W696 - 207.604 \times W2129 \\
& + 512.966 \times W396 + 106.422 \times W2239 + 79.302 \times W892 \\
& - 24.196 \times W1641 + 40.645 \times W1436 - 412.552 \times W392
\end{aligned}
\tag{4}
$$

where Ysoc is the predicted SOC, and W783, W696, W2129, W396, W2239, W892, W1641, W1436, and W392 represent the soil reflectance in 783 nm, 696 nm, 2129 nm, 396 nm, 2239 nm, 892 nm, 1641 nm, 1436 nm, and 392 nm, respectively.

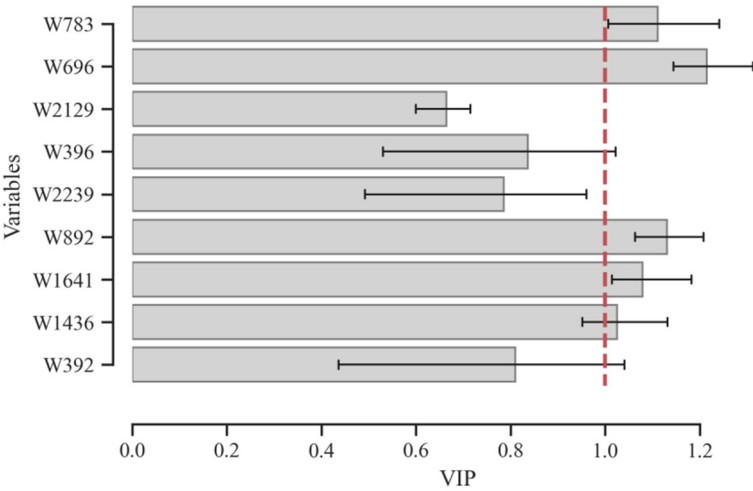

**Figure 6.** The variable importance in the projection (VIP) of SPA-PLS. Variables with VIP values greater than 1 are considered to have higher relative importance, with increasing values indicating greater significance. The red dotted line represents VIP is 1.

## 4. Discussion

### 4.1. Comparison of Different Algorithms

In this study, we found that spectral data feature selection and the construction of SOC prediction models varied greatly among different algorithms. Notably, the combination of the SPA-PLS algorithm yielded the best results.

In our study of the feature selection process for soil hyperspectral data, we found that the SPA effectively retained spectral feature information. This finding is consistent with previous studies on SOC estimation conducted by Peng et al. [18] in Jiangsu and Hubei province, and Huang et al. [33] in the Arid Area of Xinjiang. However, a study conducted in Romania found that mutual information is the optimal algorithm for screening soil spectral features [23]. One possible explanation for these differing results is that the physical structure and chemical composition of soil can vary between regions and types, leading to differences in the sensitive bands of spectral reflection and the applicability of spectral selection methods [34]. Additionally, the characteristics of the feature selection method and the number of characteristic bands may influence the feature selection process. For example, in our study, MI, R, and RF algorithms all selected bands by selecting the top ten corresponding evaluation indicators, which may have resulted in the loss of some band information. Nonetheless, considering variable complexity and information loss, SPA was determined to be the most suitable dimensionality reduction algorithm for our study.

In our study of SOC regression modeling, we found that PLS was the most suitable method. This finding is consistent with previous studies conducted by Nawar et al. [21] on salt-affected soils and Ludwig et al. [35] on an arable field in India. Some studies have indicated that RF is optimal for SOC estimation [19]. However, in our research, decision tree-based algorithms, such as GBR and RF, performed poorly. One possible explanation for this observation is that GBR and RF may exhibit inferior generalization and

predictive performance when applied to small sample datasets due to their use of bootstrap resampling to train decision trees [36]. Additionally, some studies have reported a superior performance of SVR in SOC prediction [37–39]. However, the performance of SVR may be influenced by the choice of kernel and hyperparameters, which can be challenging to optimize with small datasets.

The efficacy of feature selection and regression methods varies when used in combination. In this study, the most effective algorithmic combination was found to be SPA-PLS. This finding is consistent with previous research on soil nutrient prediction. For instance, Wang et al. [40] demonstrated that the SPA-PLS model outperformed other models in predicting rapid potassium levels in soil from Liangshan Prefecture, Sichuan province. Similarly, Morais et al. [41] and Guo et al. [42] reported that the SPA-PLS combination provided superior predictions of soil organic matter. However, Peng et al. [18] found that the SPA-SVR combination was more effective in predicting soil organic carbon levels. These discrepancies may be attributed to differences in the evaluation criteria of dimensionality reduction methods and their ability to integrate with modeling methods within the context of soil analysis, ultimately affecting model performance.

### 4.2. SOC Characteristic Wavelengths

According to the best performing SOC prediction model (SPA-PLS), the most sensitive band regions were identified at 783 nm, 696 nm, 2129 nm, 396 nm, 2239 nm, 892 nm, 1641 nm, 1436 nm, and 392 nm. However, studies on SOC sensitive bands have shown variations among different research studies. For instance, Song et al. [3] reported that characteristic wavelengths were primarily concentrated in the spectral range of 500–1000 nm and 1900–2400 nm for silty soils and 600–1400 nm and 1700–2400 nm for sandy soils. Similarly, Hu and Qi [43] found that characteristic wavelengths were mainly concentrated in the ranges of 450–619 nm, 760–909 nm, and 1968–2001 nm. These differences may be attributed to variations in soil color, texture, parent matter, and other nutrients in different regions, resulting in differences in soil spectral sensitive bands [44–46]. The absorption of water, minerals, and organic matter in soil occurs in regions of the visible near-infrared (vis-NIR) spectrum due to fundamental molecular vibrations of these substances occurring in the mid-infrared (mid-IR) range, with their overtones and combinations found in the NIR range. In the visible short-wave NIR range, electronic transitions in atoms from ground to higher energy states represent the primary mechanism by which molecules absorb energy [47]. Our results suggest that the sensitive regions identified in this study are possibly related to hydroxyl (696 nm), amine (783 nm), aromatics (1641 nm), haematite (892 nm, 396 nm, and 39 nm), carboxylic acids (143 nm), polysaccharides (2129 nm), and aliphatics (2239 nm). This indicates that the sampling soil reflectance wavelengths are primarily related to organics, based on a review of band assignments for fundamental mid-IR absorptions of soil constituents and their overtones and combinations in spectroscopy by Rossel and Behrens [22].

### 4.3. Limitations and Uncertainty

The combination of dimensionality reduction and modeling algorithms for soil hyperspectral data is a key approach to improving the prediction accuracy of SOC content [37]. In this study, we evaluated 24 combinations of 4 SOC modeling methods and 6 dimensionality reduction algorithms to identify the optimal algorithmic combination for predicting SOC based on soil hyperspectral data. Despite our efforts to optimize method combinations and spectral accuracy, several sources of uncertainty remain. As described in the Section 2 , all dimensionality reduction and modeling methods used in this study were implemented using default settings without parameter optimization, which may have limited their performance. Additionally, our optimal SOC estimation method was based on a dataset obtained using a farm-scale high-precision grid sampling approach (50 m × 50 m), which is suitable for specific sites, but may limit the regional scalability of our SOC prediction methods. Within the context of consistent small-scale management practices, spatial variations in SOC

are closely associated with environmental factors, such as soil parent material, geological conditions, and topography. However, the interactions among these environmental factors, spectral characteristics, and SOC were not explored in this study.

In light of the limitations and uncertainties outlined above, future research should aim to improve the prediction of SOC from three perspectives. First, algorithmic parameters should be optimized to identify the optimal combination of parameters for different algorithms during the modeling process. Second, the scalability of SOC prediction methods should be evaluated at small, meso-, and large scales to assess their applicability across different spatial scales. Third, remote sensing data and environmental big data should be incorporated as input variables in SOC prediction models, including long-term remote sensing data and geological data, to facilitate more convenient and rapid monitoring of SOC.

**5. Conclusions**

In this study, we evaluated the performance of various feature selection methods (RWA, R, MI, CARS, SPA, and RF) for soil hyperspectral data in combination with regression algorithms (RF, GBR, SVR, and PLS) for SOC content prediction. Among the feature selection methods, the SPA algorithm demonstrated the best performance. In terms of SOC regression algorithms, PLS and RF yielded higher $R^2$ values and a lower RMSE and MAE compared to other methods. The combination of SPA-PLS resulted in the highest $R^2$ value (0.61) and lowest RMSE (1.77 g/kg) and MAE (1.48 g/kg), indicating that SPA-PLS was the optimal algorithm combination for SOC prediction. For the optimal SOC prediction models, the relative importance of different variables in descending order was 696 nm > 892 nm > 783 nm > 1641 nm > 1436 nm > 396 nm > 392 nm > 2239 nm > 2129 nm, with 696 nm exhibiting the greatest VIP (1.22).

**Author Contributions:** Conceptualization, N.C. and Y.Z.; methodology, N.C. and X.J.; software, N.C. and X.J.; validation, N.C., X.J. and Q.L.; formal analysis, N.C. and D.C.; investigation, X.J., W.Z., D.J., X.Z. and G.D.; resources, X.J.; data curation, N.C. and X.J.; writing—original draft preparation, N.C. and X.J.; writing—review and editing, N.C. and Q.L.; visualization, N.C.; supervision, Y.Z. and Z.L.; project administration, Y.Z., Q.L. and Z.L.; funding acquisition, Q.L., Y.Z. and D.C. All authors have read and agreed to the published version of the manuscript.

**Funding:** This study was gratefully supported by the Science and Technology Project of Fujian Province (2021350000240020) and the Central Public-interest Scientific Institution Basal Research Fund (No. BSRF202103).

**Data Availability Statement:** The data presented in this study are available on request from the corresponding author. The data are not publicly available due to the project requirements.

**Conflicts of Interest:** The authors declare no conflict of interest.

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
