# Peer review of "Soil Organic Carbon Prediction Based on Different Combinations of Hyperspectral Feature Selection and Regression Algorithms"

_agronomy, doi:10.3390/agronomy13071806_

Round 1

Reviewer 1 Report

General comments:

The manuscript “Soil organic carbon prediction based on different combinations of hyperspectral dimension reduction and regression algorithms” examines the application of dimension reduction and regression algorithms for estimating SOC using hyperspectral data. The topic explored in this study is meaningful and likely to be of interest to the Agronomy community. However, I believe that the manuscript would benefit from additional clarification of the methodology, which necessitates moderate/major revisions.

Specific comments:

L44. Be specific which method can lead to toxic waste because there are different methods to test SOC. The reference provided here does not seem to be the most relevant choice.

L60-94. I think the classification between “dimension reduction” and “multivariate statistical techniques” can be problematic here. It should be noted that many dimension reduction approaches can be considered as multivariate techniques. In this context, the term "dimension reduction" primarily refers to feature selection, whereas "multivariate statistical techniques" align more accurately with regression methods, as used in the later parts of the manuscript. Certain techniques, such as RF, can fall into both categories depending on the specific method employed in the study. To address these concerns, it is essential to provide clear distinctions and utilize appropriate terminology in the revised version of the manuscript.

L203-205. Parameters used in this work should be specified, such as mtry and ntree for RF. The same applies to other methods used in this work. With current description it is difficult to tell what was done exactly and how the models were parameterized.

L220-225. Such descriptions in methods (not limited to RF) are slightly repetitive of what was already in the Introduction.

L246. This section did not make it clear whether the summary statistics were calculated through cross-validation or independent validation.

L265. Be clearer about what “stable” means.

L311-315. Such texts belong better to the Method section.

L360. it would be helpful for the manuscript to include a discussion on how the results of this work compare to previous studies. Highlighting the unique contributions of this research would provide valuable insights for the readers. Additionally, it is important to address whether the findings of this study are primarily reflective of the specific sites and environmental conditions examined or if they have broader applicability to other regions and similar applications. This information would help readers assess the generalizability and practical relevance of the study's findings.

The overall quality of the English writing is satisfactory. However, I would suggest reconsidering the use of certain words, such as "meticulously" and "bifurcating," in order to improve accessibility for a wider audience.

Reviewer 2 Report

line 127  The predominant soil type is silt loam -  it is not soil type -just soil texture class

line 140 - error within the range of 10 meters or less -  it is very high error!

Avoid repeating words meticulously

Round 2

Reviewer 1 Report

The authors did a good job with the revision.

Minor edits are recommended to improve the language of this manuscript further.